# Identifying Predictive Biomarkers for Head and Neck Squamous Cell Carcinoma Response

**DOI:** 10.3390/cancers15235597

**Published:** 2023-11-27

**Authors:** Anne-Sophie Becker, Cornelius Kluge, Carsten Schofeld, Annette Helene Zimpfer, Björn Schneider, Daniel Strüder, Caterina Redwanz, Julika Ribbat-Idel, Christian Idel, Claudia Maletzki

**Affiliations:** 1Institute of Pathology, Rostock University Medical Center, 18057 Rostock, Germany; cornelius.kluge@uni-rostock.de (C.K.); cartsen.schofeld@uni-rostock.de (C.S.); annette.zimpfer@med.uni-rostock.de (A.H.Z.); bjoern.schneider@med.uni-rostock.de (B.S.); 2Department of Otorhinolaryngology, Head and Neck Surgery “Otto Koerner”, Rostock University Medical Center, 18057 Rostock, Germany; daniel.strueder@med.uni-rostock.de; 3Department of Internal Medicine B, Cardiology, University Medicine Greifswald, 17475 Greifswald, Germany; caterina.redwanz@med.uni-greifswald.de; 4Institute of Pathology, University of Luebeck, University Hospital Schleswig-Holstein, Campus Luebeck, 23538 Luebeck, Germany; julika.ribbatidel@uni-luebeck.de; 5Department of Oto-Rhino-Laryngology & Head and Neck Surgery, University of Lubeck, University Hospital Schleswig-Holstein, Campus Luebeck, 23538 Luebeck, Germany; christian.idel@uksh.de; 6Department of Internal Medicine, Medical Clinic III—Hematology, Oncology, Palliative Medicine, Rostock University Medical Center, 18057 Rostock, Germany; claudia.maletzki@med.uni-rostock.de

**Keywords:** HNSCC, CMTM6, PD-L1, immune-checkpoint inhibitor, cisplatin, radiotherapy, tumor microenvironment

## Abstract

**Simple Summary:**

Head and neck cancer patients have a poor prognosis, even with the best therapeutic options. This is partly due to the lack of reliable prognostic biomarkers. In this study, we aimed to unravel the prognostic and predictive value of immune-related biomarkers in the tumor microenvironment. We identified the PD-L1-stabilizing and membrane-associated protein CMTM6 (CKLF-like MARVEL transmembrane domain-containing protein) as beneficial for patients receiving radiotherapy. Co-expression with the regulatory T cell transcription factor FOXP3 was predictive of a response to radiotherapy. However, no association with survival was seen for other treatment regimens, including chemotherapy, radio-chemotherapy, and immunotherapy. In the latter, CTLA-4 was associated with worse outcomes, whereas PD-L1 was not predictive. Therefore, this study describes a strategy to identify predictive markers for treatment response in head and neck cancer patients and highlights the need for treatment-specific biomarker screening.

**Abstract:**

The 5-year survival rate for head and neck squamous cell carcinoma (HNSCC) is approximately 65%. In addition to radio-chemotherapy, immunotherapy is an approach in the treatment of advanced HNSCC. A better understanding of the immune context would allow personalized treatment by identifying patients who are best suited for different treatment options. In our discovery cohort, we evaluated the expression profiles of CMTM6, PD-L1, CTLA-4, and FOXP3 in 177 HNSCCs from Caucasian patients of all tumor stages and different treatment regimens, correlating marker expression in tumor and immune cells with outcomes. Patients with CMTM6^high^-expressing tumors had a longer overall survival regardless of treatment. This prognostic benefit of CMTM6 in HNSCC was validated in an independent cohort. Focusing on the in the discovery cohort (n = 177), a good predictive effect of CMTM6^high^ expression was seen in patients receiving radiotherapy (*p* = 0.07; log rank), but not in others. CMTM6 correlated with PD-L1, CTLA-4 and FOXP3 positivity, with patients possessing CMTM6^high^/FOXP3^high^ tumors showing the longest survival regardless of treatment. In chemotherapy-treated patients, PD-L1 positivity was associated with longer progression-free survival (*p* < 0.05). In the 27 patients who received immunotherapy, gene expression analysis revealed lower levels of *CTLA-4* and *FOXP3* with either partial or complete response to this treatment, while no effect was observed for CMTM6 or PD-L1. The combination of these immunomodulatory markers seems to be an interesting prognostic and predictive signature for HNSCC patients with the ability to optimize individualized treatments.

## 1. Introduction

Head and neck squamous cell carcinoma (HNSCC) is a highly heterogeneous and common malignancy, with an increasing prevalence in the Western world. The majority of HNSCCs diagnosed worldwide are in men [1,2,3]. The major risk factors include chronic alcohol and tobacco use and high-risk human papillomavirus (HPV) infection [4,5]. The standard of care includes surgery, chemotherapy, and radiotherapy either alone or in combination with EGFR-targeting and/or immunotherapy. Although many patients initially respond well to platinum-based radio-chemotherapy, resistance and thus treatment refractoriness frequently occur. The heterogeneity of the disease and the associated individual patient response also make it difficult to standardize treatment protocols and estimate prognosis. 

To date, HPV status is the most reliable prognostic factor [6,7], but other aspects, such as molecular and pathological characteristics, tumor location, and clinical stage, additionally influence patient outcomes. The introduction of the PD1-targeting antibodies pembrolizumab and nivolumab into the first- and second-line treatment of patients with recurrent and/or metastatic (r/m) HNSCC has raised hopes for improved outcomes, but objective response rates have been reported to be less than 20%. As a result, the prognosis has not improved significantly over the decades, and the five-year overall survival rate remains at 50–60% [8,9,10,11]. 

Positivity for programmed death-ligand 1 (PD-L1) is the only approved biomarker for immune checkpoint inhibition [12,13,14]. However, the limited predictive value is a major pitfall limiting immunotherapy success. We have recently identified CKLF-like MARVEL transmembrane domain 6 (CMTM6) as a novel prognostic biomarker with, so far, unknown predictive value. CMTM6 stabilizes PD-L1 on the tumor cells’ surface by inhibiting its ubiquitination and subsequent degradation via lysosomes, exhibiting opposing functions across tumors [15,16,17,18,19]. In colorectal cancer, high CMTM6 levels are associated with an inflamed tumor microenvironment, while in hepatocellular carcinoma, CMTM6 inhibits cell proliferation by preventing p21 ubiquitination [15,20]. In HNSCC, CMTM6 was previously reported to mediate cisplatin resistance and regulate AKT-mTORC1-dependent ribosome biogenesis [21,22]. Furthermore, tumor-secreted exosomal CMTM6 was reported to induce M2-like macrophage polarization, indicating an active crosstalk between tumor and immune cells [23]. To our knowledge, studies examining the impact of CMTM6 expression in HNSCC patients receiving immunotherapy are lacking.

Given this, and the fact that the mainstay of treatment for HNSCC patients includes platinum-based (radio-)therapy or immune checkpoint inhibition, the identification of reliable predictive biomarkers remains an unmet need. Increasing evidence suggests that immune status may differ between responders and non-responders. In a very recent report, a gene signature related to the inflammatory response was described, and *OLR1* (a key receptor for oxidized low-density lipoprotein) and *INHBA* (a member of the transforming growth factor-β superfamily) were strongly associated with a poor prognosis of HNSCC patients [24]. In addition, a risk prognostic signature for predicting radiotherapy response based on the expression levels of therapeutic response-related genes has just been reported by Lin et al. [25]. In this study, we performed a detailed analysis using RNA and protein profiling of the tumor microenvironment obtained from primary and recurrent and/or metastatic HNSCC cases. We identified this marker as beneficial for patients receiving radiotherapy. Additionally, we confirmed the prognostic value of CMTM6 in an independent cohort.

## 2. Materials and Methods

### 2.1. Patient Cohort and Multi Tumor Tissue Blocks

Pretreatment samples from locally advanced and/or r/m HNSCC patients, partly treated with immune checkpoint inhibitors (ICI) (pembrolizumab, nivolumab), diagnosed between 2017 and 2021, were collected from the Rostock Pathology archives to establish our discovery cohort. If available, additional pre/post ICI treatment tumor biopsies were included. The patients’ clinicopathological characteristics, including response to treatment according to the Response Evaluation Criteria in Solid Tumors (RECIST) version 1.1 and progression-free survival (PFS), were obtained from clinical records. Using pseudonyms, follow-up data were obtained from the regional cancer registries. Tumor tissue was obtained from formalin-fixed, paraffin-embedded (FFPE) specimens. Tissue microarrays (TMAs) of three representative areas were constructed using 5 mm cores for samples from ICI-treated patients and 1 mm cores for all non-ICI-treated patients using a tissue chip microarrayer (Beecher Instruments, Silver Spring, MD, USA). The assembly of 5 mm cores preserved the tumor center and the invasion front, including infiltrating cells, for analysis and was performed with biopsy punches (Stiefel Laboratorium, Burgdorf, Germany). Tissues were collected with the patients’ consent. The institutional ethic committee at the University Hospital Rostock approved the study (number A2022-0120), which was conducted in accordance with the Declaration of Helsinki of 1975. An independent, well-described cohort from primary HNSCCs was used as a control group to validate our prognostic findings [26,27,28].

### 2.2. Gene Expression Analysis

For RNA extraction, after reviewing all original tissue slides, FFPE tissue blocks were selected and recut for hematoxylin–eosin-stained sections to determine tumor surface area. After deparaffinization, RNA was extracted from unstained FFPE tumor sections using the RNeasy FFPE kit (Qiagen, Hilden, Germany). Microdissection was performed in cases showing <50% tumor content or inked borders. mRNA expression profiling was performed using the NanoString nCounter gene expression platform (NanoString Technologies, Seattle, WA, USA). To evaluate cancer-related genes, we applied the Nanostring™ PanCancer IO 360^TM^ panel comprising 770 targets according to manufacturer’s instructions. In short, a hybridization reaction was performed using 100 ng of RNA per sample followed by further proceeding in the fully automated Prep Station with data acquisition conducted using the Digital Analyzer (performed at the Department of Internal Medicine B, Cardiology, University Medicine Greifswald, Germany). The data were exported as reporter code count (RCC) files and imported to the NanoString nSolver™ analysis software v4.0 for further analysis. Automatic quality control of mRNA was performed according to the software’s instructions.

### 2.3. Immunohistochemistry and Microscopic Evaluation

Sections of 4 μm were used. Heat-induced antigen retrieval was performed with a high pH buffer (20 min at 97 °C). The following steps were performed in an Autostainer link 48 instrument (Dako, Hamburg, Germany): 5 min of incubation in peroxidase-blocking buffer followed by 20 min of incubation with primary antibody (dilution and clone per antigen—AKT: 1:500; EPR1798, p-AKT (phosphor-S473): 1:50; EP2109Y, CMTM6: 1:1000; EPR23015-45, FOXP3: 1:200; 20034, all Abcam (Cambridge, UK)*;* cytotoxic T-lymphocyte-associated protein 4 (CTLA-4): 1:200; UMAB249 (Rockville, MD, USA), D4B9C, PDL1: 1:100; 22C3, ß2-M: 1:800, A0072; all Dako) and 3,3′- diaminobenzidine (DAB) detection using the Dako-kit K8000 according to the manufacturer’s instructions. Slides were counterstained with hematoxylin. Appropriate positive and negative controls were used. 

For assessment of immune cell distribution status (Appendix A), cytoplasmic and/or membranous reactivity of CTLA-4 and ß2 m as well as nuclear staining for FOXP3 were evaluated. Minimum and maximum expression per marker within all samples defined the ranges of the four applied categories (0: absent; 1: minimal; 2: moderate; 3: abundant), with 0/1 defined as low and 2/3 as high. Cytoplasmic and/or membranous reactivity for AKT and pAKT (S473) was scored semi-quantitatively by the percentage of positive cells and the staining intensity resulting in no/low- vs. high-expressing cases. Scores were built counting 3-5 high-power fields in the tumor center with an Axio-Cam 205 color microscope (field diameter 0.5 mm, Zeiss, Oberkochen, Germany).

PD-L1 and CMTM6 expression were scored analogous to the combined positive score (CPS) as described previously [29]. All scores were calculated within an area showing at least 100 viable tumor cells. Depending on PD-L1 and CMTM6 expression, samples were categorized as PD-L1-positive vs. -negative (cut off ≥ 1: positive) and CMTM6-low vs. -high (cut off ≥ 10: high). 

### 2.4. Statistics

Based on D’Agostino, Pearson and Shapiro–Wilk tests for normality, Pearson’s or Spearman’s correlation coefficients (r) were used to analyze the linear association between two continuous variables and used the non-parametric *t*-test Mann–Whitney tests to compare groups. Overall survival (OS) and PFS curves were constructed using the Kaplan–Meier analysis, and statistical significance was determined using the log-rank test. Unadjusted univariable analysis was conducted. All statistical tests were two-sided, and *p* values below 0.05 were considered statistically significant. All statistical analyses were performed using GraphPad^TM^ Prism^®^ v7.0 software (supplier) and nSolver analysis software (version 4.0). After removing samples with the quality control (QC) flag, mRNA count normalization and log_2_ Fold Change (FC) calculation with Welch’s *t*-test *p* values were performed. Significant DEGs were defined by *p* < 0.05 and |log_2_FC| > 1 as the threshold. 

## 3. Results

### 3.1. Patient and Tumor Characteristics 

The discovery cohort included a total of 177 patients diagnosed with r/m HNSCC with a median age of 66 years (±9.8). Clinicopathologic characteristics are shown in Table 1, where stage refers to the time of treatment initiation. Approximately 45% of all patients were smokers (≥10 py) and >32% reported regular alcohol consumption. Thus, most of the cases included here were noxae-associated, either localized in the oral cavity or oropharynx. A total of 18 patients suffered from hypopharyngeal cancer and 15 from laryngeal cancer. One regional lymph node metastasis as well as four distant metastases were obtained, while the remaining samples in the cohort consisted of locally resected primary/recurrent tumors. A total of 20% of all cases were due to a previous high-risk HPV infection.

After diagnosis, patients received either cisplatin-based chemotherapy (n = 66), radiotherapy (n = 30), combined radio-chemotherapy (n = 13), ICI (n = 27), or ICI plus chemotherapy (n = 3). A small subset of patients were treated with other agents (n = 16) or received best supportive care only due to reduced general conditions (n = 22). Median PFS on treatment was 21 months and the OS was 29 months. PFS in the ICI-treated subgroup ranged from 1 to 18 months (median 4). Using the RECIST criteria, 1 patient showed a complete response, and 10 patients showed a partial response (=responders), while 5 patients were classified as mixed response/stable disease and the remaining 10 developed tumor progression.

A validation cohort of 286 treatment-naïve primary HNSCCs derived from Caucasian patients with comparable clinicopathological parameters was included in the initial survival analysis. Anonymyzed clinical information on this patient cohort was obtained from the cooperating institute (University of Lübeck) (Figure 1). Ref. [28] provides precise information about the validation cohort. Kaplan–Meier plots for the validation cohort stratified by CMTM6 are shown in Appendix A.

### 3.2. CMTM6 Status Predicts Response to RCT but Not to Anti-PD1 Treatment

In a total of 463 patients from both cohorts combined, the mean CMTM6 CPS was 33.7 ± 32.3. PD-L1 staining was only available for the discovery cohort (n = 177), and mean PD-L1 CPS was 14.9 ± 26.2 (Table 2). Notably, 56 cases (31%) were either classified as “negative” (PD-L1 CPS < 1) or “low” (CMTM6 CPS < 10). In >40% of these cases, PD-L1 negativity correlated with low CMTM6 status.

Both markers had prognostic value, i.e., patients with PD-L1 positivity or high CMTM6 status showed a significantly better overall survival (Figure 2A,B left, *p* < 0.05). Hence, the prognostic value of PD-L1 and CMTM6 was formally confirmed in both patient cohorts. Stratification by HPV status additionally confirmed a significant prognostic impact for CMTM6 but not for PD-L1 (Figure 2A,B, middle and right). 

We then examined the predictive value of these immunologic markers in n = 177 patients from the discovery cohort (Figure 2C,D). We focused on cisplatin, RT, RCT, and ICI to cover the most relevant treatment regimens. A predictive value for PD-L1 was only seen in the cisplatin subgroup. Here, patients with a CPS ≥ 1 showed a significantly better OS than their counterparts with CPS < 1 (Figure 2B). No statistically significant benefit was seen for the other regimens. For CMTM6, there was a trend towards a better outcome when the tumor was classified as high and the patient received RT (*p* = 0.07). Likewise, a slightly improved overall survival was seen in the other treatment groups, i.e., cisplatin or RCT. However, patients with high PD-L1 and/or CMTM6 tumors who received anti-PD1 ICI treatment had a comparable OS to their counterparts (Figure 2B,C). Hence, no predictive impact of either marker was seen in this subgroup.

### 3.3. Correlation of CMTM6 Status with Other Immune Biomarkers

The interplay between different immune cell subtypes within the tumor microenvironment (TME) prompted us to check for additional biomarkers with potential prognostic and predictive value (Spearman correlation) in the discovery cohort. For this analysis, treatment-unrelated (=prognostic) and treatment-related (=predictive) correlations were conducted (Table 3). For the latter, only patients who received cisplatin, RT, or RCT were included. CMTM6 was analyzed for correlation with PD-L1, CTLA-4, FOXP3, AKT, and pAKT. Table 2 and Table 3 show the staining results and the correlations between the markers. Table 4 refers to staining pattern depending on the clinical parameters.

Independent of treatment, CMTM6 expression strongly correlated with FOXP3, a specific transcriptional regulator in Tregs. Moderate correlations were observed between CMTM6 and PD-L1 and CTLA-4. Notably, a strong correlation between CMTM6 and FOXP3 was seen across treatments. In all subgroups, CMTM6 correlated more strongly with CTLA-4 status than PD-L1. No association was found between CMTM6 and AKT or pAKT.

Then, the prognostic role of these immune markers was analyzed (Figure 3). High FOXP3 expression levels were associated with significantly improved overall survival in both HPV^pos^ and HPV^neg^ cases (Figure 3A, *p* < 0.05). To study if FOXP3 has comparable prognostic relevance to CMTM6, the overall survival of patients was analyzed in additional subgroups. Therefore, tumors were classified into FOXP3^low^/CMTM6^low^ (n = 51), FOXP3^high^/CMTM6^low^ (n = 6), FOXP3^low/^CMTM6^high^ (n = 17), and FOXP3^high^/CMTM6^high^ (n = 99). Patients in the latter group and those with FOXP3^high^/CMTM6^low^ tumors had the best prognosis with survival rates exceeding 50% at the longest time point. This underlines the prognostic impact of both markers in HNSCC.

To see if this prognostic role could be transferred to the predictive model, we performed a subgroup analysis with respect to the individual treatments. A trend towards a better outcome was observed for all treatment regimens (Figure 3B), reaching statistical significance for patients receiving RCT (*p* < 0.05 vs. FOXP3^low^/CMTM6^low^). A closer look at the survival curve of ICI-treated patients revealed a delayed favorable response with a median OS of 43 months in the FOXP3^high^/CMTM6^high^ subgroup (vs. 22 months FOXP3^low^/CMTM6^low^). Thus, although statistical significance was not achieved in this small patient cohort, we confirmed the predictive value by doubling the median OS of patients with r/m HNSCC. 

CTLA-4 had no effect on median OS. Finally, the relevance of serine/threonine protein kinase AKT and its active phosphorylated form (pAKT) was analyzed, because AKT is a downstream target of CMTM6-mediated Wnt signaling and may be involved in cisplatin resistance. Hence, AKT is considered as a proto-oncogene. In our study, we identified a slightly improved median OS in the AKT high subgroup (vs. AKT low) and vice versa when pAKT was considered. Still, no significant survival benefit was seen in either group. 

Further treatment-related analysis was only performed for the cisplatin subgroup but did not yield any prognostic value. Hence, the prognostic value of AKT/pAKT in our patient cohort is rather low.

### 3.4. Immunotherapy Response in r/m HNSCC Gene Signatures in Responders

Transcriptomic and immunological profiling was performed on pre- and post-treatment samples of r/m HNSCC patients receiving anti-PD1 immunotherapy. These samples were from the discovery cohort and only included one patient with high-risk oncogenic HPV infection; the remaining samples were HPV-negative. Gene expression analysis for anti-tumor immune activity showed two distinct subgroups (Figure 4A). Concerning the profiled gene signatures, responders had significantly lower mRNA expression levels of CTLA-4 than non-responders (Figure 4B). By plotting the receiver operator characteristics (ROCs), CTLA-4 had the highest capacity to predict response (area under the curve 0.753; PD-L1: 0.7) of 43 signatures. While the expression levels of genes associated with T cells, including exhausted CD8^+^ cytotoxic cells, as well as levels of the immune checkpoint receptor T cell immunoglobulin and ITIM domain (TIGIT) were lower in responders, this failed to be statistically significant in the multivariate analysis (Figure 4C). In line with our findings described above, the abundance of Tregs as measured by gene expression of FOXP3 was lower in these immunologically cold tumors, here showing a better treatment response to ICI, and vice versa. Patients benefiting longer from ICI had lower expression genes levels in tumors involved in signatures of CD8 T cells, TIGIT, and CTLA-4 compared to patients with shorter progression times (Figure 4D). Interestingly, we found no differences in PD-L1 expression between responders and non-responders. Genes involved in the pathway downstream of the IFN gamma axis were significantly higher in samples defined as “PD-L1-positive” by IHC (Figure 4E). In three patients with re-biopsy material at the time point of progression during ICI treatment, neither CMTM6 nor PD-L1 status by IHC changed according to the applied cut-offs, and Beta-2-microglobulin expression was preserved. Focusing on the profiled immune cell population in these patients (one classified as a responder, two as non-responders), the tumor inflammation signature (TIS), which predicts a response to pembrolizumab, decreased in the two non-responders (Figure 4F). On the other hand, CTLA-4 levels increased in the third patient, who progressed after 8 months of a partial response (Figure 4G). We also focused on the seven included patients with available tumor material (a) either before or (b) at disease progression during ICI treatment. Using IHC, PD-L1, CMTM6, FOXP3 and CTLA-4 status was stable in two patients with mixed changes in the remaining patients. Immune profiling generally revealed inhibition of immune cell abundance and antitumor immune activity; however, an increase in inhibitory immune signaling was observed when primary tumors were compared with the material from r/m disease, as previously described [28].

## 4. Discussion

The identification of reliable biomarkers for HNSCC is ongoing. In this study, we applied a classical pathological approach and examined the TME of HPV-related and -unrelated cases to elucidate the biological relevance of several known and novel biomarkers. We included cases from different sites and treatment regimens to cover the clinical heterogeneity and related challenges. The overall outcome was comparable to reports in the literature and confirmed the beneficial response to combined radio-chemotherapy compared to either monotherapy in the first-line treatment of HNSCC. However, in the r/m setting, the use of ICIs has little impact on PFS or OS with frequent relapse in all patients. 

In our discovery cohort, we observed that CMTM6 positivity correlates with an improved OS, independently of adjuvant treatment modalities. We observed this beneficial prognostic impact in an independent validation cohort from treatment-naïve HNSCC patients with comparable clinical characteristics. CMTM6 is now well known to stabilize PD-L1 protein expression on the tumor cells’ surface [19,30]. It is widely expressed on both tumor and immune cells and plays an important role in regulating T cell activation and antitumor response. The prognostic impact of CMTM6 is comparable to PD-L1, underlying its overlapping biological functions. However, the predictive value of this novel biomarker is largely unknown [31]. By stratifying patients from the discovery cohort according to the most frequent treatment regimens, i.e., chemotherapy, radiotherapy (RT), radio-chemotherapy, or immune checkpoint inhibition, a predictive value was only seen in the RT group. Patients with high CMTM6 expression on tumor cells and/or in the TME showed a survival benefit compared to their CMTM6-low counterparts. While HPV is an approved biomarker, the prognostic impact was even higher here when tumors were also CMTM6-high. The prognostic impact of CMTM6 is in accordance with reports on colorectal (CRC) and hepatocellular carcinomas [15,20] but in contrast to the findings of Chen et al. [32,33]. In their cohort of 210 primary HNSCC samples collected from the School and Hospital of Stomatology, Wuhan University, China, the authors identified CMTM6 overexpression by IHC as a poor prognostic marker. Using digital image analysis, membranous and/or cytoplasmic CMTM6 positivity in tumor cells predicted a shorter OS independent of other clinicopathological parameters. Their scoring system for CMTM6, analogous to the tumor proportion score (TPS), excluded immune cells surrounding the tumor. In our study, we quantified CMTM6 positivity using the CPS. This score includes cells within the TME, which regulates tumor cell survival and confers anticancer effects [3,32]. Although the group describes similar staining patterns for PD-L1 and CMTM6 as seen here, the results are not comparable due to the different scoring methods. Furthermore, Chen et al. performed their study in an Asian cohort, where risk factors and outcomes differ from those in Caucasians [33]. With this in mind, the authors reviewed The Cancer Genome Atlas (TCGA) [34] to support their findings. While they do not provide information on the impact of *CMTM6* RNA expression in the TCGA cohort, they show that both *CMTM6* RNA expression and protein levels, when analyzed analogously to TPS, increased with higher pathological grade. Our cohort includes 21% HPV-related and therefore conventionally ungraded tumors. We did not see any correlation between tumor grade and CMTM6. However, another Chinese group investigated *CMTM6* RNA levels in CRC samples and found a higher *CMTM6* expression in early clinical stages (UICC I/II vs. III/IV) [15] (ref. Peng). Although CRC as an adenocarcinoma is not comparable to HNSCC, this underlies the importance of future studies to elucidate the impact of CMTM6 in solid cancers. 

Still, no such correlation was seen in the other subgroups. The most unexpected finding was the missing predictive value for the ICI subgroup, which is described for lung carcinoma [35]. The same missing correlation was seen for PD-L1. Here, a beneficial effect was only obtained when patients received cisplatin but not with other therapies. Although this finding is somehow unexpected, it fits with recent reports in which ICI-based treatments yielded mixed responses and sometimes failed to improve the outcome of r/m HNSCC patients [36,37,38]. Hence, PD-L1 and/or CMTM6 alone may be insufficient to predict ICI responses in this patient cohort.

Here, we performed a more global approach by combining protein and gene expression-based analysis in patients from the discovery cohort. Firstly, we identified a strong correlation between CMTM6 and FOXP3. The latter is a specific transcription factor in regulatory T cells with known immunosuppressive effects and a correspondingly conflicting role in patient prognosis. In HNSCC, high FOXP3 levels have previously been reported to correlate with better outcomes [39,40] regardless of treatment [41]. Here, we further confirmed the prognostic impact independent of HPV status. Comparable results were shown in gastric cancer, where co-expression of FOXP3 and CMTM6 was related to a favorable prognosis [42]. FOXP3 was found to co-localize with tumor-infiltrating CD20^+^CD27^+^ B cells to promote effector and memory T cell differentiation and enhance B cell and NK cell activation and function. Therefore, the dual role of FOXP3 in cancer is advantageous for HNSCC patients, and this positive impact is probably enhanced by CMTM6. Adding to this, Jing et al. recently described an inflammatory response-related gene profile to predict the prognosis in HNSCC [24]. In a parallel study, a risk model was developed to predict therapy response and prognosis in response gene-defined subgroups of HNSCC [25]. With our study, we suggest additional predictive biomarkers that may be potential targets in the near future.

Another novel finding of our study was the strong correlation between CMTM6 and CTLA-4, which was even stronger than with PD-L1. This is intriguing and describes a new, previously unknown interaction of CMTM6 with immunoregulatory molecules [19]. Unlike CMTM6, CTLA-4 was not expressed in tumor cells, as reported recently by Hoffmann et al. [43]. In this study, lower CTLA4 promoter methylation correlated with a response to ICI and longer progression-free survival, but these findings were not confirmed at the protein level. This adds to Sholl’s conclusion that not only protein but also genomic and transcriptomic levels mediate the variable response to ICIs [44].

The interaction between the tumor and host immune-specific genetic and epigenetic factors is complex and it is now clear that PD-L1 protein expression by immunohistochemistry is neither sensitive nor specific enough to serve as a reliable predictive biomarker. Our study adds to this by showing that protein and gene expression levels of PD-L1 were insufficient predictors of a response to immunotherapy. Beyond PD-L1, the transcriptomic profile of this entity can identify possible correlations between an ICI response, TME, and CMTM6. By means of gene expression analysis, we found that the mRNA levels of *CTLA-4* and *FOXP3* were lower in the responder group of our patients treated with ICI. Since CTLA-4 is mainly expressed in tumor-draining lymph nodes, established tissue-based biomarkers for predicting its inhibitory capacity are lacking. In melanoma patients, the anti-CTLA-4 inhibitor ipilimumab is the standard of care regardless of PD-L1 status, and the combination with an anti-PD1 ICI improves its efficacy. Although some studies have failed to show a beneficial effect of the Nivo/Ipi combination, several phase II/III trials are currently investigating the efficacy of anti-CTLA-4 antibodies as a monotherapy or in combination with anti-PD1 antibodies in HNSCC (ClinicalTrials.gov Identifiers: NCT04080804, NCT04326257, NCT03624231, NCT03212469, NCT03799445). In addition, a lower CD8 profile indicated a trend towards an improved ICI response, which is in contrast to the data from Ayers et al. [45]. Using mRNA expression profiling, they showed that a T cell-inflamed microenvironment characterized by active IFN-γ signaling, cytotoxic effector molecules, and T cell-active cytokines is a common feature of the biology of tumors that respond to PD1 checkpoint blockade. With this knowledge, the authors established a focused set of genes to identify this PD1 checkpoint blockade-responsive biology. Although the genes analyzed were congruent with those studied here, possible explanations for the different findings may be that (1) our study cohort included less patients (n = 26 here vs. n = 43 by Ayers et al.), (2) only n = 8 patients received ICI as a first-line treatment, (3) the PFS in our cohort was shorter, and (4) PD-L1 CPS dis not predict patients’ responsiveness to treatment. 

Combination therapies targeting PD1 and CTLA-4 may also fail due to primary or secondary resistance mechanisms. Beta-2-microglobulin (ß2M) is a major histocompatibility complex class I component and is essential for antigen presentation. Its loss, either by mutation or degradation, leads to ICI treatment failure [46,47]. Similarly, all tumors in the responder group expressed ß2M, whereas three non-responders lacked this protein on the tumor cell surface prior to treatment initiation. The three samples with re-biopsy material at the time of progression during ICI treatment had all retained ß2M expression, supporting alternative resistance mechanisms. 

Finally, the relevance of the serine/threonine kinase AKT and its active form, pAKT, was studied. In oral cancer, pAKT is overexpressed because of reduced protein degradation and contributes to tumor growth, lymph node metastasis, and shorter survival time [48,49]. Other studies related pAKT expression to radio-resistance [50] or HPV positivity [51]. In our study, neither AKT nor its active form were associated with a significant survival benefit. This finding is rather expected and hence additional studies on larger patient cohorts will have to clarify the relevance of AKT/pAKT in HNSCC.

A limitation of our study was the small sample size of only n = 26 HNSCCs in the ICI cohort for gene expression profiling. However, as anti-PD1 treatment had only recently been approved for the treatment of HNSCC, the present cohort is of reasonable size, including eight patients without prior systemic therapy. Second, the TMA design weakens our results because separation into the tumor center, invasion front, and stromal compartment is more difficult than on whole mount sections. To address this, we used 1 mm cores in the construction of the TMA and punched tumor tissue multiple times to capture the heterogeneity of the TME. In addition, the biomarkers CMTM6 and PD-L1 were examined on whole slides. We also extracted mRNA from whole tumors, including immune cells. To increase the tumor cell content and decrease bystander cells, tumor microdissection was performed. In challenging cases, we could have used laser microdissection to separate tumor and immune cells, but the risk of losing mRNA-containing cells by focusing on the small amount of tumor in some of our cases seemed too high. Next, the measurement of CMTM6 was not well defined. To include CMTM6 expression on tumor and associated immune cells, we scored CMTM6 analogously to PD-L1 by calculating its CPS, which itself has its weaknesses. Receiver operating curve analysis revealed a cut-off of ≥10 for CMTM6. Future studies in independent cohorts will show whether another scoring system for CMTM6 protein expression is more valuable. Additionally, fluorescent labeling of the examined molecules, including immune cells, would prove the co-expression of CMTM6 with other markers in more detail. While membranous co-localization of CMTM6 and PD-L1 was shown via high-resolution florescence several times [35,52], the association with other co-stimulatory molecules, including FOXP3 and CTLA-4, especially in HNSCC, is less clear. Therefore, this question should be addressed in a future project, maybe even incorporating other co-stimulatory surface molecules like CD28.

## 5. Conclusions

In Caucasian HNSCC patients, CMTM6^high^ tumors showed a longer OS in independent cohorts. CMTM6 correlates with PD-L1-, CTLA-4-, and FOXP3-positive Treg infiltration in HNSCCs with a better response to radiotherapy in CMTM6^high^ tumors. While neither PD-L1 nor CMTM6 status predicted the treatment response to immune checkpoint inhibitors, and low CTLA-4 expression was associated with a good response. This argues for the inclusion of CTLA-4 inhibitors to target the antitumor immunity in HNSCC patients.

## Figures and Tables

**Figure 1 cancers-15-05597-f001:**
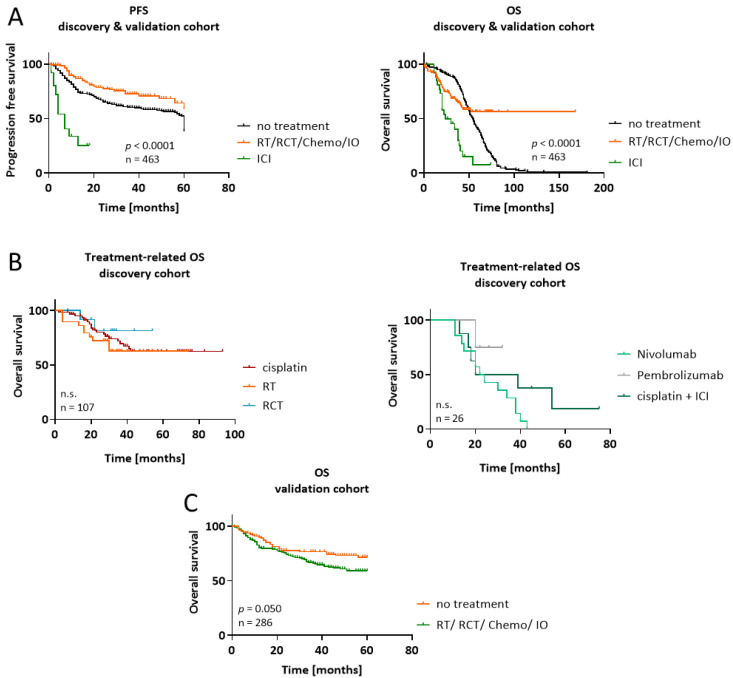
Kaplan–Meier survival curves of the HNSCC patients. The discovery (n = 177) and validation cohort (n = 286) included HNSCC patients with comparable clinicopathological parameters receiving first- or second-line treatment or best supportive care. (**A**) Progression-free survival (PFS) and overall survival (OS) for the patients from both cohorts. Treatment regimens included radiotherapy (RT), radio-chemotherapy (RCT), chemotherapy, immune-oncology (IO), or ICI (immune checkpoint inhibition). (**B**) Treatment-related OS in the discovery cohort (n = 177) stratified according to the specific regimen. (left) Cisplatin-based chemotherapy (n = 64), RT (n = 29) or RCT (n = 13). (right) ICI mono (n = 14 and 4, respectively) or ICI-based combination regimens (n = 8). (**C**) Treatment-related OS in the validation cohort (n = 286). Log-rank analysis was performed to study differences between the individual regimens. n.s.—not significant.

**Figure 2 cancers-15-05597-f002:**
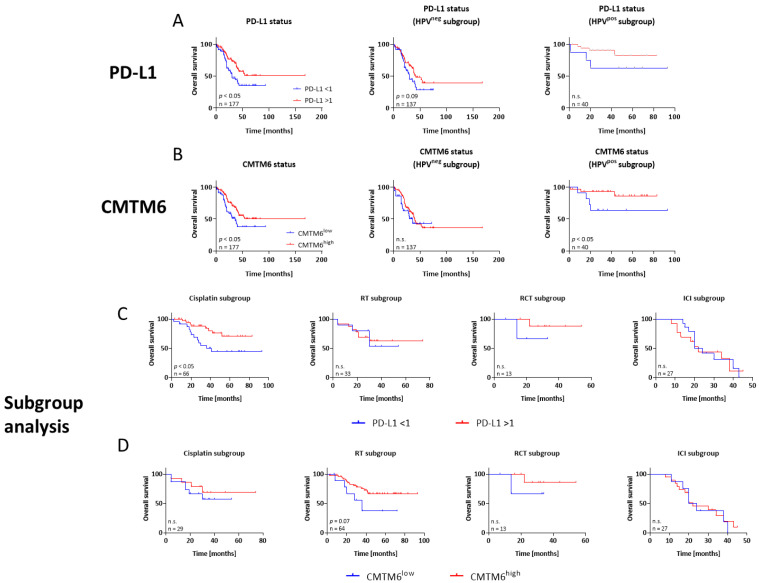
Kaplan–Meier survival curves of HNSCC patients. For the biomarker-driven and treatment-related prognosis, analysis was performed on n = 177 patients (discovery cohort) receiving first- or second-line treatment. Treatment regimens included chemotherapy, radiotherapy (RT), radio-chemotherapy (RCT), or ICI (immune checkpoint inhibition). (**A**,**B**) Overall survival (OS) according to PD-L1 or CMTM6 in all patients (left) and stratified according to HPV status (middle and right). (**C**) Treatment-related OS stratified according to the specific regimen and PD-L1 status. (**D**) Treatment-related OS stratified according to the specific regimen and CMTM6 status. Numbers of patients for each treatment group, i.e., cisplatin, RT, RCT, or ICI are given in their respective graphs. Log-rank analysis was performed to study any differences between the individual regimens. n.s.—not significant.

**Figure 3 cancers-15-05597-f003:**
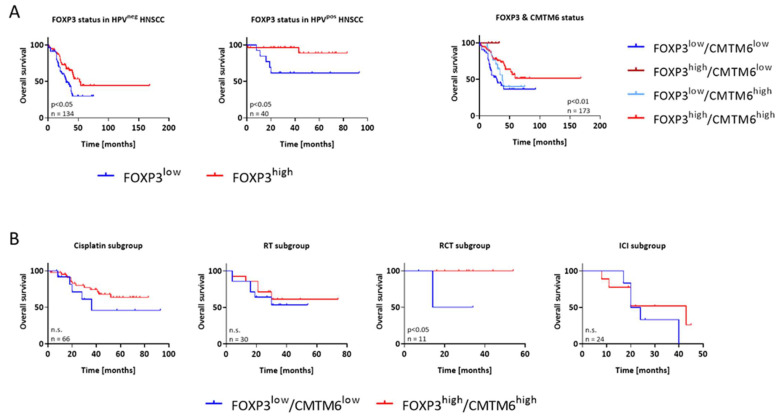
Kaplan–Meier survival curves of HNSCC patients from the discovery cohort. For biomarker-driven and treatment-related prognosis, analysis was performed on n = 177 patients receiving first- or second-line treatment. Treatment regimens included chemotherapy, radiotherapy (RT), radio-chemotherapy (RCT), or ICI (immune checkpoint inhibition). (**A**) Overall survival (OS) of patients according to FOXP3 and combined FOXP3/CMTM6 status. (**B**) Treatment-related OS stratified according to the specific regimen and FOXP3/CMTM6 status. Cisplatin-based chemotherapy, RT, RCT, or ICI. Log-rank analysis was performed to study any differences between the individual regimens. n.s.—not significant.

**Figure 4 cancers-15-05597-f004:**
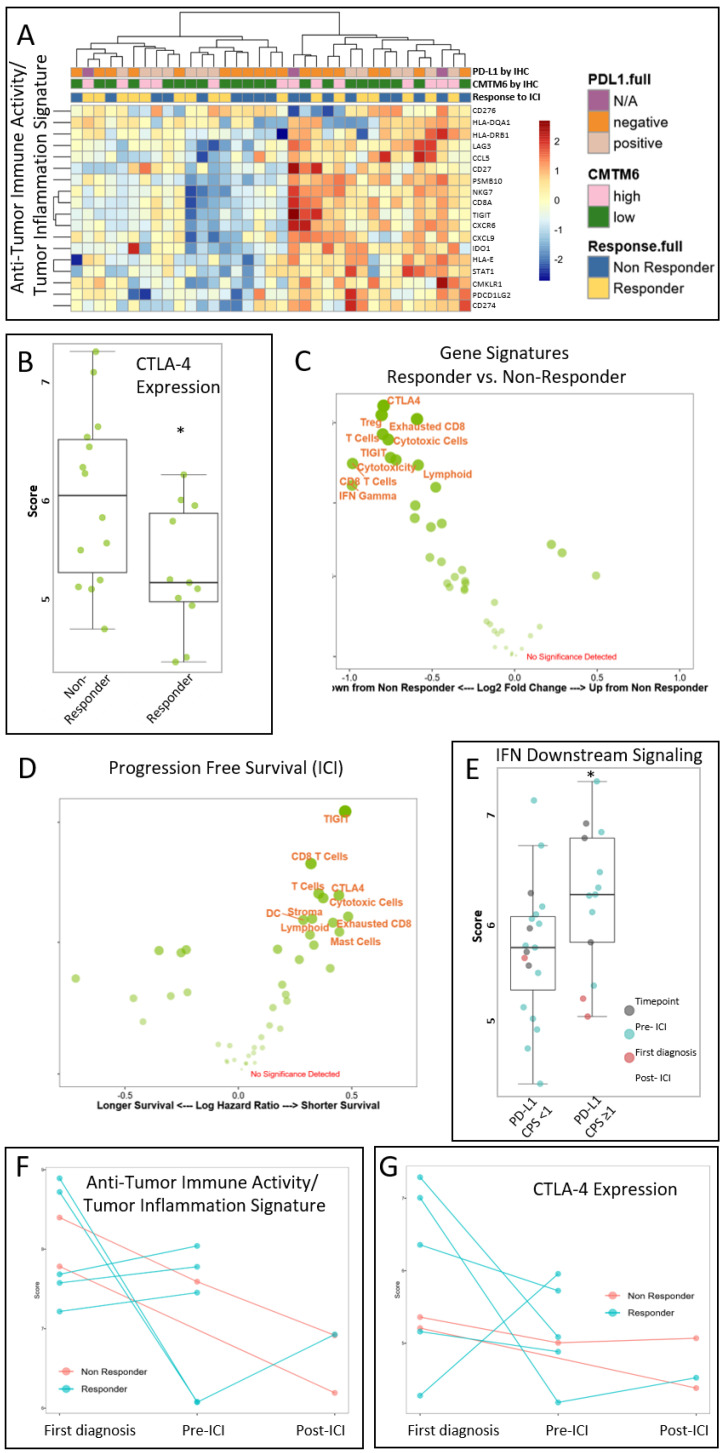
Gene signatures in HNSCC patients from the discovery group with their response to immune checkpoint inhibitors. (**A**) Heatmap representing the down- and upregulated genes involved in antitumor immune activity by unsupervised hierarchical clustering. Scores are scaled to have a mean of zero and a standard deviation of one. The standardized signature scores are truncated at ±3 standard deviations (99% of the data should fall within ±3 standard deviations of the mean). (**B**) Box plot indicating a significantly lower CTLA-4 signature in patients with a response to ICI as assessed by univariate analysis (*p* = 0.02). (**C**) Volcano plot of all gene signatures displays no significant difference between the ICI responders and non-responders in a multivariate analysis. Up indicates upregulated and down indicates downregulated genes. (**D**) Survival volcano plot of all gene signatures displays no significant difference between the signature’s hazard ratios in a multivariate analysis. Signatures further to the right are associated with a decreased risk of an event relative to the baseline. (**E**) Box plot indicating a significantly higher IFN downstream signature in patients whose tumors showed a CPS ≥ 1 for PD-L1 immunohistochemistry as assessed by univariate analysis (*p* = 0.04). (**F**) Spaghetti plot displaying a decrease in antitumor immune activity from initial diagnosis to r/m disease with start of ICI until the time point of progression under ICI in non-responders, and (**G**) an increase in CTLA-4 in recurrent tumor material from a patient who initially had a partial response to ICI. * *p* < 0.05. CPS—combined positive score; Pre-ICI—sample taken just before the start of ICI; Post-ICI—sample taken just after progression under ICI.

**Table 1 cancers-15-05597-t001:** Clinico-pathological characteristics of the discovery cohort.

Group Characteristics	Σ n = 177
**Female n (%)** **Male n (%)**	34 (19.2) 143 (80.8)
**Median age (years ± SD)**	66 ± 9.8
**Noxae** smoking (>10 py in %): yes/no alcohol (>1 drink/d in %): yes/no no information (smoking/alcohol)	51.9/34.5 36.7/63.3 11.3/2.3
**p16/HPV status (%)** positive negative	21 79
**Tumor stage (UICC stage 8th edition) (n)** 1/2/3/4	37/56/29/55
**Localization (n)** oral cavity oropharynx hypopharynx larynx	79 65 18 15
**Treatment (n)** C/RT/RCT/ICI/ICI + C Other/no adjuvant therapy	66/33/13/27/3 16/19
**Median survival (month)** PFS OS	21.0 ± 20.6 29.0 ± 21.4

Values are given as absolute/relative numbers and mean ± SD unless otherwise stated. Abbreviations: py—pack years; d—day; HPV—human papilloma virus; RT—radiotherapy; RCT—radio-chemotherapy; C—chemotherapy (not specified); ICI—immune checkpoint inhibition (nivolumab or pembrolizumab); PFS—progression-free survival; OS—overall survival.

**Table 2 cancers-15-05597-t002:** Immunohistochemistry staining results for the markers in the discovery group.

	Defined Criteria for Evaluation			
	Cut-Off *	Low/Negative	High/Positive	Median
n = 177	n (%)	n (%)	n (%)	n (%)
**CMTM6**	(CPS) 10	57 (32.2)	120 (67.8)	33.7
**PD-L1**	(CPS) 1	58 (32.8)	119 (67.2)	14.9
**CTLA-4**	(density of positive cells) *low* vs. *high*	93 (52.5)	84 (47.5)	-
**FOXP3**	(density of positive cells) *low* vs. *high*	77 (43.5)	100 (56.5)	-
**AKT**	(% positive cells × staining intensity) *no/low* vs. *high*	119 (67.2)	58 (32.8)	-
**pAKT**	(% positive cells × staining intensity) *no/low* vs. *high*	152 (85.9)	25 (14.1) 67.2	-

* in (brackets): scheme for evaluation. Abbreviations: CPS, combined positive score.

**Table 3 cancers-15-05597-t003:** Spearman correlation for markers in the discovery cohort including samples from patients receiving no adjuvant or either chemotherapy/radiation/chemoradiation.

CMTM6 vs.		Biomarker				
		PD-L1	CTLA-4	FOXP3	AKT	pAKT
Treatment Unrelated n = 151 ^#^	r	0.26	0.63	0.85	0.31	0.22
	95% CI	0.098 to 0.41	0.52 to 0.72	0.80 to 0.89	0.15 to 0.46	0.047 to 0.38
	*p* (two-tailed)	0.0014	<0.0001	<0.0001	0.0002	0.0110
	*p* value summary	**	****	****	***	*
RT n = 33	r	0.32	0.76	0.89	0.19	0.16
	95% CI	−0.061 to 0.62	0.54 to 0.88	0.78 to 0.95	−0.20 to 0.54	−0.26 to 0.53
	*p* (two-tailed)	0.0877	<0.0001	<0.0001	0.3221	0.4350
	*p* value summary	ns	****	****	ns	ns
Cisplatin n = 66	r	0.36	0.55	0.85	0.03	0.13
	95% CI	−0.12 to 0.56	0.34 to 0.70	0.76 to 0.91	−0.25 to 0.31	−0.16 to 0.40
	*p* (two-tailed)	0.0037	<0.0001	<0.0001	0.8311	0.3636
	*p* value summary	**	****	****	ns	ns
RCT n = 13	r	0.51	0.84	0.71	0.29	0.31
	95% CI	−0.10 to 0.85	0.53 to 0.95	0.25 to 0.91	−0.39 to 0.77	−0.38 to 0.77
	*p* (two-tailed)	0.089	0.0006	0.0078	0.4182	0.3549
	*p* value summary	ns	***	**	ns	ns

Interpretation: ≥0.70: very strong relationship; 0.40–0.69: strong relationship; 0.30–0.39: moderate relationship; 0.20–0.29: weak relationship; 0.01–0.19: no relationship. ns—not significant. ^#^ PD-L1 n = 148; AKT n = 134; pAKT n = 130. * *p* < 0.05; ** *p* < 0.01; *** *p* < 0.001; **** *p* < 0.0001.

**Table 4 cancers-15-05597-t004:** Contingency table for the markers in the discovery group.

	Biomarker											
n = 177	CMTM6 (n)			PD-L1 (n)			FOXP3 (n)			CTLA-4 (n)		
	Low	High	*p*	Negative	Positive	*p*	Low	High	*p*	Low	High	*p*
Gender			ns			ns			ns			ns
Male	38	105		48	95		60	83		75	68	
Female	9	25		10	24		17	17		18	16	
Smoking ^#^			ns			ns			ns			ns
Yes	33	59		38	54		37	55		56	36	
No	29	36		20	45		35	30		33	32	
Alcohol ^##^			ns			ns			ns			ns
Yes	30	31		28	33		24	37		28	33	
No	43	69		41	71		49	63		65	47	
HPV												
Pos.	10	28		9	29		13	26		15	24	
Neg.	47	92		48	91		64	74		69	69	
TNM			ns			ns			ns			*
T1	9	28		12	25		18	19		10	27	
T2	18	38		15	41		26	30		25	31	
T3	11	18		13	16		9	20		15	14	
T4	19	36		18	37		24	31		34	21	
N−/N+	23/34	66/54	ns	23/35	66/53	ns	37/40	48/52	ns	40/44	49/44	ns
M0/M1	43/14	102/17	ns	43/15	103/16	ns	63/14	83/17	ns	65/19	81/12	ns
Recurrence												
No	34	92	*	24	91	****	48	91	****	25	100	**
Yes	23	28		34	28		27	9		21	31	

^#^ for n = 157; ^##^ for n = 173; * *p* < 0.05; ** *p* < 0.01; **** *p* < 0.0001. Abbreviations: *p*—*p* value; HPV—human papilloma virus; ns—not significant.

## Data Availability

The data presented in this study are available on request from the corresponding author.

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
