# Peer review of "Identifying Predictive Biomarkers for Head and Neck Squamous Cell Carcinoma Response"

_cancers, 2023, doi:10.3390/cancers15235597_

Round 1
Reviewer 1 Report (Previous Reviewer 2)
Comments and Suggestions for Authors
I reviewed the manuscript and believe it is now worthy of publication.
Author Response
Dear reviewer,
we thank you for the assesment that the manuscript is ready for publication.
Reviewer 2 Report (New Reviewer)
Comments and Suggestions for Authors
The Manuscripts “Identifying Predictive Biomarkers for Head and Neck Squamous Cell Carcinoma Response” is a well-structured manuscript that provides valuable insights into the role of immune-related biomarkers in the prognosis and treatment of head and neck squamous cell carcinoma (HNSCC).The manuscript is well-organized, and the language is clear. The title accurately reflects the content of the study. The methods section could benefit from more detailed information about the patient cohort, such as demographic data. The results are interesting and potentially significant. The finding that CMTM6high expressing tumors had a longer overall survival regardless of treatment is particularly noteworthy. Especially, its prediction in response to RCT, but not to anti-PD1 treatment and correlation with other immune biomarkers. The discussion could benefit from a more in-depth analysis of how these findings compare to previous research in the field. Additionally, discussing potential mechanisms behind these observations would strengthen the paper.
Overall, this is a promising study that contributes valuable knowledge to the field of HNSCC research. With some minor revisions, this manuscript has the potential to make a significant impact. The manuscript should be accepted with minor revision.
1. The figure as tables should be formatted and clearly labelled as the text is not properly visible in Figures: e.g., Figure 4 A gene expression data etc.
2. It would be good to add abbreviations of genes etc. used in the paper to be listed in the manuscript.
Author Response
Dear reviewer, we would like to express our gratitude for taking the time to review our manuscript and for providing us with valuable feedback. We appreciate your comments and suggestions, which have helped us to improve the manuscript. We have addressed the major issues raised and updated the manuscript accordingly. Please see below our detailed responses for each point, with changes in the revised manuscript highlighted in yellow (except for the figure).
Response to reviewer 2 minor points:
- The figure as tables should be formatted and clearly labelled as the text is not properly visible in Figures: e.g., Figure 4 A gene expression data etc.
We appreciate this comment and formatted the tables and figures more properly. Following your suggestion, we particularly enlarged the axis and scale labels of Figure 4 and adapted the sub-headings (4A: gene expression analysis; 4C: CTLA-4 Gene Signature; 4D: Progression Free Survival). Please see the updated version of the manuscript for details.
- It would be good to add abbreviations of genes etc. used in the paper to be listed in the manuscript.
We thank the reviewer for this good advice and included an abbreviation list at the end of the manuscript (page 18).
We are very confident that the enhanced version of the manuscript now matches the requirements for publication in Cancers. We would be very pleased if you would find this enhanced version suitable for publication.
This manuscript is a resubmission of an earlier submission. The following is a list of the peer review reports and author responses from that submission.
Round 1
Reviewer 1 Report
Comments and Suggestions for Authors
The authors Anne-Sophie Becker et al, in the manuscript entitled “Identifying Predictive Biomarkers for Head and Neck Squamous Cell Carcinoma Response”, presents results from IHQ and gene expression analysis of CMTM6 and other related markers in FFPE samples from 177 HNSCC patients with different treatment settings.
The presentation of the manuscript results should be improved by adding a table where markers CPS and other measures are summarized. This information might be more relevant than the correlations presented in table 2. Authors should decide whether this table be included in the main text or as supplementary data.
Table 1 should be revised since information about the number of patients or percentages of them is missing for some categories. Also, “noxae” category does not add up to 100% of the samples.
In figure 1 authors state that a comparison is made between their cohort and another from the University of Lübeck, but results shown are from the addition of both cohorts. If both cohorts are added in the survival analysis then no comparison can be made. This controversy should be resolved.
A paper by Lei Chen (Chen, Lei, Qi-Chao Yang, Yi-Cun Li, Lei-Lei Yang, Jian-Feng Liu, Hao Li, Yao Xiao, Lin-Lin Bu, Wen-Feng Zhang, and Zhi-Jun Sun. 2020. “Targeting CMTM6 Suppresses Stem Cell-Like Properties and Enhances Antitumor Immunity in Head and Neck Squamous Cell Carcinoma.” Cancer Immunology Research 8 (2): 179–91.) shows results from CMTM6 expression in HNSCC by IHQ. These authors found that CMTM6 overexpression in HNSCC is associated with a poor prognosis. Please include this paper in the manuscript discussion section with your comments on their findings.
In the introduction section from line 87 to 94 three sentences describe recent findings related to other tumor markers in HNSCC. Authors should consider commenting on these papers in the discussion section.
A supplementary table is mentioned in the manuscript but is lacking in the online version at the susy portal.
Comments on the Quality of English Language
There are some minor issues like the absence of definition for the acronym “TME” (although known by many, it would be better to present the phrase properly), and also check for bi-omarkers hyphenation.
Reviewer 2 Report
Comments and Suggestions for Authors
This study looked for possible prognostic immune-related biomarkers to help decide the best-suited treatment regimen for individuals with head and neck cancer. Despite the smaller sample size, this paper provides avenues for further investigations in the field, which is beneficial as these cancer patients generally have a low survival rate. The main comment is that the authors need to make it clear across the entire paper the number of patients on which they are establishing their conclusions.
As such, the paper could be improved by addressing the following:
Page 5, line 196: The authors mentioned a control group of 286 patients who did not receive any treatment; they are showing their survival rate for these patients in Figure 1 but do not provide any characteristics for these patients or analysis of the biomarkers. Are the characteristics of this group of patients similar (Nbr of female, male, age, etc…) to those of the cohort they are studying? How are the biomarkers changing with the disease progression is this control group?
Page 7, line 307: Make clear that in this case the entire cohort is (n = 177).
In Figure 2, precise the n value for PD-L1<1, PD-L1>1, CMTM6 low, and CMTM6 on the graphs.
Page 8, line 337: Does the title should not be "Correlation of CMTM6 with other immune biomarkers"?
Additionally, in this section, mention that you are referring to Table 2.
In table 2: add the number of patients in each group (e.g., treatment unrelated, RT, etc…)
Page 9, line 362-363: Please provide the number in each group.
Reviewer 3 Report
Comments and Suggestions for Authors
The authors used literature review to identify potential prognostic/predictive markers in head neck squamous cell carcinoma and performed immunohistochemistry, largely of TMAs on a cohort of 177 mixed HPV-positive and HPV-negative as well as recurrent metastatic and untreated locally aggressive tumors to correlate with survival. The markers examined by immunohistochemistry were: AKT, phospho-AKT, CMTM6, FOXP3, CTLA-4, and PD-L1. Expression of these markers was correlated with overall survival and some markers were correlated with overall survival after specific treatments of chemotherapy, radiation therapy, radiochemotherapy, and ICI therapy. PD-L1 CPS greater than or equal to 1 and CMTM6 CPS greater than or equal to 10 correlated with overall survival in this cohort. Subgroup analysis suggested a trend to survival improvement in the high CMTM6 group treated with radiation. In patients whose tumors had PD-L1 CPS greater than or equal to 1 overall survival was improved for those treated with cisplatin. No correlation with either marker for overall survival was seen with immune checkpoint therapy. High FOXP3 expression was associated with improved overall survival in patients with high FOXP3 and high CMTM6 expression showed the best survival when examining the entire cohort. Tumors with high FOXP3 and high CMTM6 compared to tumors with low expression for both markers had improved survival when treated with radiochemotherapy and longer survival when treated with immune checkpoint inhibition.
Gene expression analysis was performed to identify markers correlating with overall survival in patients treated with immune checkpoint inhibition. These analyses revealed that CTLA-4 expression was lower in patients who responded to therapy.
Although the findings are potentially interesting relating to FOXP3 and CMTM6, as well as PD-L1 expression as prognostic biomarkers, it is hard to imagine how this will or can translate to practice. These markers, with the questionable exception of the combination of FOXP3 and CMTM6, did not correlate with response to immune checkpoint therapy as might be expected for immune markers. Although response to standard therapies could also relate to immune activity, there was no strong or consistent correlation of these markers with any particular treatment type, suggesting that tumors with these markers may respond better to all therapies. The mixed nature and small size of the cohort makes it very difficult or impossible to parse more details about response to different therapies. A major weakness of the study is not separating tumors by HPV status, given that HPV is a major driver of prognosis and response to therapy. In addition, markers should be assessed by TNM classification and recurrence status to determine if they improve prognostic power compared to staging or apply to recurrent metastatic tumors.
A minor point relates to the cutpoint of CMTM6 which was determined by receiver operator curves according to the manuscript, but this data is not included or referenced.